# The Structural Identifiability of a Humidity-Driven Epidemiological Model of Influenza Transmission

**DOI:** 10.3390/v14122795

**Published:** 2022-12-15

**Authors:** Chunyang Zhang, Xiao Zhang, Yuan Bai, Eric H. Y. Lau, Sen Pei

**Affiliations:** 1School of Mathematics, Changchun Normal University, Changchun 130032, China; 2WHO Collaborating Centre for Infectious Disease Epidemiology and Control, School of Public Health, Li Ka Shing Faculty of Medicine, The University of Hong Kong, Hong Kong, China; 3Laboratory of Data Discovery for Health Limited, Hong Kong Science and Technology Park, Hong Kong, China; 4Department of Environmental Health Sciences, Mailman School of Public Health, Columbia University, New York, NY 10032, USA

**Keywords:** influenza, structural identifiability, climate-driven epidemiological model, scaling method

## Abstract

Influenza epidemics cause considerable morbidity and mortality every year worldwide. Climate-driven epidemiological models are mainstream tools to understand seasonal transmission dynamics and predict future trends of influenza activity, especially in temperate regions. Testing the structural identifiability of these models is a fundamental prerequisite for the model to be applied in practice, by assessing whether the unknown model parameters can be uniquely determined from epidemic data. In this study, we applied a scaling method to analyse the structural identifiability of four types of commonly used humidity-driven epidemiological models. Specifically, we investigated whether the key epidemiological parameters (i.e., infectious period, the average duration of immunity, the average latency period, and the maximum and minimum daily basic reproductive number) can be uniquely determined simultaneously when prevalence data is observable. We found that each model is identifiable when the prevalence of infection is observable. The structural identifiability of these models will lay the foundation for testing practical identifiability in the future using synthetic prevalence data when considering observation noise. In practice, epidemiological models should be examined with caution before using them to estimate model parameters from epidemic data.

## 1. Introduction

Influenza epidemics cause considerable morbidity and mortality every year worldwide [1,2,3,4,5,6]. According to The World Health Organization, influenza leads to about 3 to 5 million cases of severe illness and about 290,000 to 650,000 deaths annually [7]. This burden might be alleviated by understanding historical and current transmission dynamics and predicting further transmission trends of influenza to assist public health authorities in designing effective interventions and vaccination strategies, especially in some special situations (for example, the emergence of influenza A (H1N1) viruses and the potential rebound of influenza in the post-COVID-19 pandemic period) [8,9,10,11,12].

Climate-driven epidemiological models, such as models driven by humidity, are mainstream tools for understanding seasonal transmission dynamics and predicting future trends in influenza activity. Recently, climate-driven epidemiological models have been successfully applied to recreate historical activity time series of influenza and to forecast the week with the highest influenza activity in temperate, tropical and subtropical regions (for example, in the United States and Hong Kong) [13,14,15,16,17]. Confident predictions using these models depend on testing various numerical optimization algorithms, such as filter-based data-assimilation algorithms, which fit the model to epidemic data when parameterizing models to simulate the transmission dynamics of influenza [13]. However, the model structure identifiability needs to be tested to avoid the optimization algorithm falling into a set of the locally optimal solution.

Model identifiability includes structural and practical identifiability, involving investigation of whether unknown model parameters can be uniquely determined from noise-free epidemic data and accurately identified from noisy data, respectively [18,19,20,21,22,23,24]. The structural identifiability of the model is a fundamental prerequisite for practical identifiability and for the model to be used in practice [25]. It is necessary, but insufficient, to accurately identify model parameters from actual noisy data because a model that is structurally identifiable might be unidentifiable when noisy data are used. The scaling method, which is a structural identifiability analysis method that has been proposed in recent years, is based on the scale invariance of the equations [26]. Compared with existing structural identifiability methods (such as differential algebra), this method has the advantage of simple operation (no advanced computing skills are required) and low computational cost, particularly when analyzing high-dimensional non-linear models [27,28,29,30]. This method has been used to analyze the structural identifiability of mathematical modeling describing biological processes, such as the generalized mass-action model [31,32,33].

In this study, we apply the scaling method to analyse the structural identifiability of several types of commonly used humidity-driven epidemiological models. We investigate whether the key epidemiological parameters (infectious period, the average duration of immunity, the average latency period, the maximum and minimum daily basic reproductive number) can be determined simultaneously when the population prevalence of infected people is observable.

## 2. Methods

Here, we briefly introduce the process from building an epidemiological model to applying the model in real-world applications. After building an epidemiological model of influenza transmission, we test the structural identifiability of the model to investigate the properties of the model itself (the upper part of Figure 1). If the model is structurally identifiable, then we test the practical identifiability of the model, which determines whether the model is identifiable for noise data (e.g., reported noise), using synthetic data experiments. If the model is also practically identifiable, then this model can potentially be used in practice after evaluating its performance for specific functions (e.g., inference, forecasting, etc.). On the other hand, if a model is structurally unidentifiable, any parameter estimated by optimization algorithms might be unreliable; then, we need to consider modifying the model. For some complex models, for example, agent-based influenza transmission models [34], we can only test the practical identifiability directly after building the model (the lower part of Figure 1), as current mathematical methods may not be able to theoretically test the structural identifiability of these complex models. Here, we mainly focus on testing the structural identifiability of humidity-driven epidemiological models.

### 2.1. Humidity-Driven Epidemiological Model

We test the structural identifiability of several commonly used humidity-driven epidemiological models. The form of each model is as follows.
1.SIS model:
(1)$$\begin{array}{cc} \frac{dS}{dt} & =\frac{I}{D}-\frac{\beta (t)IS}{N}-\alpha ,\\ \frac{dI}{dt} & =\frac{\beta (t)IS}{N}-\frac{I}{D}+\alpha . \end{array}$$2.SIRS model:
(2)$$\begin{array}{cc} \; & \frac{dS}{dt}=\frac{N-S-I}{L}-\frac{\beta (t)IS}{N}-\alpha ,\\ \; & \frac{dI}{dt}=\frac{\beta (t)IS}{N}-\frac{I}{D}+\alpha . \end{array}$$3.SEIR model:
(3)$$\begin{array}{cc} \frac{dS}{dt} & =-\frac{\beta (t)IS}{N}-\alpha ,\\ \frac{dE}{dt} & =\frac{\beta (t)IS}{N}-\frac{E}{W}+\alpha ,\\ \frac{dI}{dt} & =\frac{E}{W}-\frac{I}{D}. \end{array}$$4.SEIRS model:
(4)$$\begin{array}{cc} \frac{dS}{dt} & =\frac{N-S-E-I}{L}-\frac{\beta (t)IS}{N}-\alpha ,\\ \frac{dE}{dt} & =\frac{\beta (t)IS}{N}-\frac{E}{W}+\alpha ,\\ \frac{dI}{dt} & =\frac{E}{W}-\frac{I}{D}. \end{array}$$
where *N*, *S*, *E*, and *I* represent the total number of people, the number of susceptible people, the number of exposed people, and the number of infectious individuals, respectively. N=S+I in the SIS model, N=S+I+R ie all parameters and unobse N=S+E+I+R in the SEIR model and the SEIRS model. α represents the rate at which influenza viruses are imported into the model due to travel. *t* represents time, such as the day, week or year. β(t) represents the transmission rate at time t. *D* represents the mean infectious period in all four models. *L* represents the average duration of immunity in the SIRS model and the SEIRS model. *W* represents the average latency period in the SEIR model and the SEIRS model. The flow diagrams of these models are presented in Figure A1 in Appendix A.

The basic reproductive number, $R_{0}\left(t\right)$, represents the average number of secondary infections generated by a primary case in a fully susceptible population at time *t*, which is proportional to the transmission rate. The expression is as follows:(5)$$R_{0}\left(t\right)=\beta \left(t\right)D.$$

The influenza virus survival and transmission are relative to the absolute humidity (AH) shown from laboratory experiments [35]. The specific humidity (SH) is a measure of AH, in which q(t) represents SH at time *t*. In this model, the humidity factor modulates $R_{0}\left(t\right)$ through an exponential relationship:(6)$$R_{0}\left(t\right)=e^{\left(a\cdot q(t)+b\right)}+R_{0\min},$$
where a=−180 is estimated by fitting laboratory influenza virus survival to the value of AH using a regression model. $\;b=\log(R_{0\max}-R_{0\min})$, $\;R_{0\max}$ and $R_{0\min}$ are the maximum and minimum daily basic reproductive number, respectively. Parameter sets $\Theta_{1}=\left\{D,R_{0\max},R_{0\min}\right\}$ in the SIS model, $\Theta_{2}=\left\{D,L,R_{0\max},R_{0\min}\right\}$ in the SIRS model, $\Theta_{3}=\left\{D,W,R_{0\max},R_{0\min}\right\}$ in the SEIR model and $\Theta_{4}=\left\{D,L,W,R_{0\max},R_{0\min}\right\}$ in the SEIRS model may be estimated by fitting the model to the epidemic data. We analyse the structural identifiability of these models, which tests whether each parameter set can be uniquely determined simultaneously when the prevalence data is observable (using the observation record from the beginning to the end time, such as one influenza season in temperate regions).

### 2.2. The Frameworks for the Scaling Method

The scaling method is easy to use for identifying the structural identifiability of a non-linear model based on simple scaling transformations and the solution of simple sparse systems of equations [26]. The ordinary differential equations(ODE) model, which is applied to the frameworks of the scaling method, is as follows:(7)$$\begin{array}{cc} \; & \frac{dx_{i}}{dt}=f_{i}\left(x_{1},\cdots ,x_{s},x_{s+1},\cdots ,x_{n};\theta_{1},\cdots ,\theta_{m}\right),\\ \; & x_{i}\left(0\right)=x_{i,0},\;\;\;i=1,\cdots ,n, \end{array}$$
where $x_{i}\left(0\right)$ represents the initial conditions, $\frac{dx_{i}}{dt}$ represents the change of $x_{i}$ over time, depending on *m* parameters $\theta_{j}$, and the number of state variables is *n*. $f_{i}$ is a function characterising the specific details of the change rate of $x_{i}$. The simplicity of this method depends on the ability to decompose functions $f_{i}$ as a sum of *P* functional independent components, $f_{ij}$,
(8)$$\begin{array}{cc} \frac{dx_{i}}{dt} & =f_{i}\left(x_{1},\cdots ,x_{s},x_{s+1},\cdots ,x_{n};\theta_{1},\cdots ,\theta_{m}\right)\\ \; & =\sum_{j=1}^{P}f_{ij}\left({\bar{x}}_{j},{\bar{\theta }}_{j}\right). \end{array}$$

A property of $f_{ij}$ is that $f_{ij}$ is functionally independent of $f_{ik}$ for every j≠k. Here, $\bar{x_{j}}$ and $\bar{\theta_{j}}$ represent the subset of variables and parameters of function $f_{ij}$, respectively. In simple terms, if $f_{i}\left(x_{i},x_{2},\ldots \right),\ldots ,f_{n}\left(x_{i},x_{2},\ldots \right)$ are linearly independent functions, then the only solution of the equation
(9)$$\sum_{i=1}^{n}a_{i}f_{i}\left(x_{1},x_{2},\ldots \right)=0$$
is $a_{1}=a_{2}=,\ldots ,=a_{n}=0$. The functional independence theorems used in this work are presented in Appendix A.

Based on decomposing functions, the steps of the scaling method are as follows:Step 1.Scale all parameters and unobserved variables using unknown scaling factors, μ:
(10)$$\begin{array}{cc} \; & \theta_{i}\to \mu_{\theta_{i}}\theta_{i}\;\;\;i=1,\cdots ,m,\\ \; & x_{j}\to \mu_{x_{j}}x_{j}\;\;\;j=s+1,\cdots ,n, \end{array}$$
and substitute them into Equation (Equation 8). The experiment measures variables $x_{1},\ldots ,x_{s}$ without modifying them (*n* is the total number of state variables, $x_{1},\ldots ,x_{s}$ is the observable state variable and $x_{s+1},\ldots ,x_{n}$ is the unobserved state variable).Step 2.Obtain the scaled version for each functionally independent function. Namely,
(11)$$f_{ij}\left({\bar{x}}_{j},{\bar{\theta }}_{j}\right)=f_{ij}\left({\bar{x}}_{j},\mu_{{\bar{\theta }}_{j}}{\bar{\theta }}_{j}\right)\;\;\;i=1,\cdots ,s$$
and
(12)$$f_{ij}\left({\bar{x}}_{j},{\bar{\theta }}_{j}\right)=\frac{1}{\mu_{x_{i}}}f_{ij}\left(\mu_{{\bar{x}}_{j}}{\bar{x}}_{j},\mu_{{\bar{\theta }}_{j}}{\bar{\theta }}_{j}\right)\;\;\;i=s+1,\cdots ,n.$$Step 3.Find scaling factor combinations that maintain the system invariant. Only the parameters ${\bar{\theta }}_{j}$ with a solution $\mu_{{\bar{\theta }}_{j}}=1$ are identifiable. Only the variables, ${\bar{x}}_{j}$ with $\mu_{{\bar{x}}_{j}}=1$ are observable. Otherwise, parameters whose scaling factors are coupled form identifiable groups but cannot be identified independently.

## 3. Results

Here, we demonstrate how to use the simple scaling method to test the identifiability of the four humidity-driven epidemiological models introduced in the Methods section.

We consider a scenario where only *I* is observed, representing a kind of epidemic data that can be collected in practice. For example, in the UK, the COVID-19 Infection Survey identified those people testing positive for coronavirus (COVID-19) in private residential households (surveillance sensors) at a point in time to help the government make decisions on how to respond to the emerging epidemic and provide information to the public [36]. This infection survey can, in principle, be extended to survey influenza to identify new positive cases of influenza regularly around the influenza season. In this scenario, we can obtain *I* from the sentinel surveillance systems. We test whether the humidity-driven epidemiological models in Equations (Equation 1)–(Equation 4) are structurally identifiable, respectively.

### 3.1. SIS Model

For the SIS model, we test whether parameter set $\Theta_{1}=\left\{D,R_{0\max},R_{0\min}\right\}$ can be determined uniquely from the observable *I*. First, we investigate whether the differential equation in Equation (Equation 1) can be decomposed into a sum of linearly independent functions. This is a prerequisite for using the scaling method. For the differential equation associated with *S*, we have:(13)$$\begin{array}{c} f_{S1}=\frac{I}{D},\;f_{S2}=-\frac{\beta (t)IS}{N}. \end{array}$$

According to Theorem A1, the generalized Wronskian determinant is as follows:(14)$$W_{S}=\left|\begin{array}{cc} {▵}_{0}f_{S1} & {▵}_{0}f_{S2}\\ {▵}_{1}f_{S1} & {▵}_{1}f_{S2} \end{array}\right|=\left|\begin{array}{cc} \frac{I}{D} & -\frac{\beta (t)IS}{N}\\ \frac{1}{D} & -\frac{\beta (t)(S+I)}{N} \end{array}\right|=-\frac{\beta \left(t\right)I^{2}}{DN}\neq 0.$$

So, $f_{S1}$ and $f_{S2}$ are linearly independent functions. Similarly, for the differential equation associated with *I*, we have:(15)$$\begin{array}{c} f_{I1}=\frac{\beta (t)IS}{N},\;f_{I2}=-\frac{I}{D}. \end{array}$$

The corresponding generalized Wronskian determinant is as follows:(16)$$W_{I}=\left|\begin{array}{cc} {▵}_{0}f_{I1} & {▵}_{0}f_{I2}\\ {▵}_{1}f_{I1} & {▵}_{1}f_{I2} \end{array}\right|=\left|\begin{array}{cc} \frac{\beta (t)IS}{N} & -\frac{I}{D}\\ \frac{\beta (t)(S+I)}{N} & -\frac{1}{D} \end{array}\right|=\frac{\beta \left(t\right)I^{2}}{DN}\neq 0.$$

So, $f_{I1}$ and $f_{I2}$ are linearly independent functions. Next, we explore whether parameter set $\Theta_{1}=\left\{D,R_{0\max},R_{0\min}\right\}$ can uniquely be determined from the observable *I* using the scaling method. The steps of the scaling method are as follows:Step 1.We scale the parameter set $\Theta_{1}=\left\{D,R_{0\max},R_{0\min}\right\}$ and the unobserved variable (*S*) by unknown scaling factors:
(17)$$\left\{\begin{array}{c} D\to \mu_{D}D,\\ R_{0\max}\to \mu_{R_{0\max}}R_{0\max},\\ R_{0\min}\to \mu_{R_{0\min}}R_{0\min},\\ S\to \mu_{S}S. \end{array}\right.$$Step 2.We obtain the scaled version for each functional linear independent function in Equations (Equation 13) and (Equation 15).Step 3.We obtain the identifiability equations:
(18)$$\begin{array}{c} \frac{I}{\mu_{S}\mu_{D}D}=\frac{I}{D}, \end{array}$$
(19)$$\begin{array}{cc} \; & \frac{I\mu_{S}S}{\mu_{S}N\mu_{D}D}\left[e^{aq(t)}\left(\mu_{R_{0\max}}R_{0\max}-\mu_{R_{0\min}}R_{0\min}\right)+\mu_{R_{0\min}}R_{0\min}\right]\\ \; & \;\;=\frac{IS}{ND}\left[e^{aq(t)}\left(R_{0\max}-R_{0\min}\right)+R_{0\min}\right], \end{array}$$
(20)$$\begin{array}{cc} \; & \frac{I\mu_{S}S}{N\mu_{D}D}\left[e^{aq(t)}\left(\mu_{R_{0\max}}R_{0\max}-\mu_{R_{0\min}}R_{0\min}\right)+\mu_{R_{0\min}}R_{0\min}\right]\\ \; & \;\;=\frac{IS}{ND}\left[e^{aq(t)}\left(R_{0\max}-R_{0\min}\right)+R_{0\min}\right], \end{array}$$
(21)$$\begin{array}{c} \frac{I}{\mu_{D}D}=\frac{I}{D}. \end{array}$$

Manipulating the above formulas, the identifiability equations are:(22)$$\left\{\begin{array}{c} \mu_{D}=1,\\ \mu_{S}=1,\\ e^{aq(t)}\left(\mu_{R_{0\max}}R_{0\max}-\mu_{R_{0\min}}R_{0\min}\right)+\mu_{R_{0\min}}R_{0\min}\\ \;=e^{aq(t)}\left(R_{0\max}-R_{0\min}\right)+R_{0\min}. \end{array}\right.$$

From the last formula in Equation (Equation 22), we further manipulate this formula and obtain the following equation:(23)$$\begin{array}{c} e^{aq(t)}\left[\left(\mu_{R_{0\max}}-1\right)R_{0\max}-\left(\mu_{R_{0\min}}-1\right)R_{0\min}\right]=\left(1-\mu_{R_{0\min}}\right)R_{0\min}. \end{array}$$
$e^{aq(t)}$ is not equal to zero. When the left and right sides of the equation are equal, we have:(24)$$\left\{\begin{array}{c} \left(\mu_{R_{0\max}}-1\right)R_{0\max}-\left(\mu_{R_{0\min}}-1\right)R_{0\min}=0,\\ \left(1-\mu_{R_{0\min}}\right)R_{0\min}=0,R_{0\max}>0,R_{0\min}>0. \end{array}\right.$$

By solving the above equation, we can obtain:(25)$$\left\{\begin{array}{c} \mu_{R_{0\max}}=1,\\ \mu_{R_{0\min}}=1. \end{array}\right.$$

Therefore, the SIS model is identifiable. Namely, parameter set $\Theta_{1}=\left\{D,R_{0\max},R_{0\min}\right\}$ can be determined uniquely from the observable *I*.

### 3.2. SIRS Model

For the SIRS model, we test whether the parameter set $\Theta_{2}=\left\{D,L,R_{0\max},R_{0\min}\right\}$ can be determined uniquely from the observable *I*. First, we investigate whether the differential equation in Equation (Equation 2) can be decomposed into a sum of linearly independent functions. For the differential equation associated with *S*, we have: For the differential equation associated with *S*, we have:(26)$$\begin{array}{cc} \; & f_{S1}=\frac{N-S-I}{L},f_{S2}=-\frac{\beta (t)IS}{N}. \end{array}$$

According to Theorem A1, the generalized Wronskian determinant is as follows:(27)$$W_{S}=\left|\begin{array}{cc} {▵}_{0}f_{S1} & {▵}_{0}f_{S2}\\ {▵}_{1}f_{S1} & {▵}_{1}f_{S2} \end{array}\right|=\left|\begin{array}{cc} \frac{N-S-I}{L} & -\frac{\beta (t)IS}{N}\\ \frac{-2}{L} & -\frac{\beta (t)S}{N}-\frac{\beta (t)I}{N} \end{array}\right|=\frac{\beta (t)[I(I-N)+S(S-N)]}{LN}\neq 0.$$

So, $f_{S1}$ and $f_{S2}$ are linearly independent functions. Similarly, for the differential equation associated with *I*, we have:(28)$$\begin{array}{c} f_{I1}=\frac{\beta (t)IS}{N},\;f_{I2}=-\frac{I}{D}. \end{array}$$

The corresponding generalized Wronskian determinant is as follows:(29)$$W_{I}=\left|\begin{array}{cc} {▵}_{0}f_{I1} & {▵}_{0}f_{I2}\\ {▵}_{1}f_{I1} & {▵}_{1}f_{I2} \end{array}\right|=\left|\begin{array}{cc} \frac{\beta (t)IS}{N} & -\frac{I}{D}\\ \frac{\beta (t)(S+I)}{N} & -\frac{1}{D} \end{array}\right|=\frac{\beta \left(t\right)I^{2}}{DN}\neq 0.$$

So, $f_{I1}$ and $f_{I2}$ are linearly independent functions. Next, we explore whether the parameter set $\Theta_{2}=\left\{D,L,R_{0\max},R_{0\min}\right\}$ can be determined uniquely from the observable *I* using the scaling method. The steps of the scaling method are as follows:Step 1.We scale the parameters and unobserved variables by unknown scaling factors:
(30)$$\begin{array}{c} \left\{\begin{array}{c} D\to \mu_{D}D,\\ L\to \mu_{L}L,\\ R_{0\max}\to \mu_{R_{0\max}}R_{0\max},\\ R_{0\min}\to \mu_{R_{0\min}}R_{0\min},\\ S\to \mu_{s}S. \end{array}\right. \end{array}$$Step 2.We obtain the scaled version for each functional linear independent function in Equations (Equation 26) and (Equation 28).Step 3.We obtain the identifiability equations:
(31)$$\frac{N-\mu_{S}S-I}{\mu_{S}\mu_{L}L}=\frac{N-S-I}{L},$$
(32)$$\begin{array}{c} \frac{I\mu_{S}S}{\mu_{S}\mu_{D}DN}\left[e^{aq(t)}\left(\mu_{R_{0\max}}R_{0\max}-\mu_{R_{0\min}}R_{0\min}\right)+\mu_{R_{0\min}}R_{0\min}\right]\\ =\frac{IS}{DN}\left[e^{aq(t)}\left(R_{0\max}-R_{0\min}\right)+R_{0\min}\right], \end{array}$$
(33)$$\begin{array}{c} \frac{I\mu_{S}S}{\mu_{D}DN}\left[e^{aq(t)}\left(\mu_{R_{0\max}}R_{0\max}-\mu_{R_{0\min}}R_{0\min}\right)+\mu_{R_{0\min}}R_{0\min}\right]\\ =\frac{IS}{DN}\left[e^{aq(t)}\left(R_{0\max}-R_{0\min}\right)+R_{0\min}\right], \end{array}$$
(34)$$\frac{I}{\mu_{D}D}=\frac{I}{D}.$$

Manipulating Equations (Equation 31)–(Equation 34), the identifiability equations are:(35)$$\left\{\begin{array}{c} \mu_{D}=1,\\ \mu_{S}=1,\\ \mu_{L}=1,\\ \left(\mu_{R_{0\max}}-1\right)R_{0\max}=\left(\mu_{R_{0\min}}-1\right)\left(1-\frac{1}{e^{aq(t)}}\right)R_{0\min} \end{array}\right.$$

From the last formula in Equation (Equation 35), the left side of Equation (Equation 35) is constant, the right side of this equation is the function of *t*, and $e^{aq(t)}$ is not equal to zero. For the equation to be satisfied, $\mu_{R_{0\max}}-1=0$ and $\mu_{R_{0\min}}-1=0$. Hence, the system has a unique solution ($\mu_{R_{0\max}}=\mu_{R_{0\min}}=1$). It follows that the SIRS model is identifiable.

### 3.3. SEIR Model

For the SEIR model, we test whether the parameter set $\Theta_{3}=\left\{D,W,R_{0\max},R_{0\min}\right\}$ can be determined uniquely from the observable *I*. First, we investigate whether the differential equation in Equation (Equation 3) can be decomposed into a sum of linearly independent functions. For the differential equation associated with *S*, we have:(36)$$\begin{array}{c} f_{S1}=-\frac{\beta (t)IS}{N}. \end{array}$$

According to Theorem A1, the generalized Wronskian determinant is as follows:(37)$$\begin{array}{c} W_{S}=\left|\begin{array}{c} {▵}_{0}f_{S1} \end{array}\right|=-\frac{\beta (t)IS}{N}\neq 0. \end{array}$$

So, $f_{S1}$ is a linearly independent function. Similarly, for the differential equation associated with *E*, we have:(38)$$f_{E1}=\frac{\beta (t)IS}{N},\;f_{E2}=-\frac{E}{W}.$$

The corresponding generalized Wronskian determinant is as follows:(39)$$W_{E}=\left|\begin{array}{cc} {▵}_{0}f_{E1} & {▵}_{0}f_{E2}\\ {▵}_{1}f_{E1} & {▵}_{1}f_{E2} \end{array}\right|=\left|\begin{array}{cc} \frac{\beta (t)IS}{N} & -\frac{E}{W}\\ \frac{\beta (t)(S+I)}{N} & -\frac{1}{W} \end{array}\right|=\frac{\beta (t)}{NW}\left[E(S+I)-IS\right]\neq 0.$$

So, $f_{E1}$ and $f_{E2}$ are linearly independent functions. For the differential equation associated with *I*, we have:(40)$$\begin{array}{c} f_{I1}=\frac{E}{W},\;f_{I2}=-\frac{I}{D}. \end{array}$$

The generalized Wronskian determinant is as follows:(41)$$W_{I}=\left|\begin{array}{cc} {▵}_{0}f_{I1} & {▵}_{0}f_{I2}\\ {▵}_{1}f_{I1} & {▵}_{1}f_{I2} \end{array}\right|=\left|\begin{array}{cc} \frac{E}{W} & -\frac{I}{D}\\ \frac{1}{W} & -\frac{1}{D} \end{array}\right|=\frac{I-E}{WD}\neq 0.$$

So, $f_{I1}$ and $f_{I2}$ are linearly independent functions. Next, we explore whether parameter set $\Theta_{3}=\left\{D,W,R_{0\max},R_{0\min}\right\}$ can be determined uniquely from observable *I* using the scaling method. The steps of the scaling method are as follows:Step 1.We scale the parameter set $\Theta_{3}=\left\{D,W,R_{0\max},R_{0\min}\right\}$ and the unobserved variable (*S*) by unknown scaling factors:
(42)$$\left\{\begin{array}{c} W\to \mu_{W}W,\\ D\to \mu_{D}D,\\ R_{0\max}\to \mu_{R_{0\max}}R_{0\max},\\ R_{0\min}\to \mu_{R_{0\min}}R_{0\min},\\ S\to \mu_{S}S,\\ E\to \mu_{E}E. \end{array}\right.$$Step 2.We obtain the scaled version for each functional linear independent function in Equations (Equation 36), (Equation 38) and (Equation 40).Step 3.We obtain the identifiability equations:
(43)$$\begin{array}{cc} \; & \frac{I\mu_{S}S}{\mu_{S}N\mu_{D}D}\left[e^{aq(t)}\left(\mu_{R_{0\max}}R_{0\max}-\mu_{R_{0\min}}R_{0\min}\right)+\mu_{R_{0\min}}R_{0\min}\right]\\ \; & \;\;=\frac{IS}{ND}\left[e^{aq(t)}\left(R_{0\max}-R_{0\min}\right)+R_{0\min}\right], \end{array}$$
(44)$$\begin{array}{cc} \; & \frac{I\mu_{S}S}{\mu_{E}N\mu_{D}D}\left[e^{aq(t)}\left(\mu_{R_{0\max}}R_{0\max}-\mu_{R_{0\min}}R_{0\min}\right)+\mu_{R_{0\min}}R_{0\min}\right]\\ \; & \;\;=\frac{IS}{ND}\left[e^{aq(t)}\left(R_{0\max}-R_{0\min}\right)+R_{0\min}\right], \end{array}$$
(45)$$\begin{array}{c} \frac{\mu_{E}E}{\mu_{E}\mu_{W}W}=\frac{E}{W}, \end{array}$$
(46)$$\begin{array}{c} \frac{\mu_{E}E}{\mu_{W}W}=\frac{E}{W}, \end{array}$$
(47)$$\begin{array}{c} \frac{I}{\mu_{D}D}=\frac{I}{D}. \end{array}$$

According to formulas (Equation 45)–(Equation 47), it is easy to get:(48)$$\left\{\begin{array}{c} \mu_{W}=1,\\ \mu_{D}=1,\\ \mu_{E}=1. \end{array}\right.$$

Pressing $\mu_{D}=1$ into the formula (Equation 43), we can use Equation (Equation 23) in the SIS model. Then by Equation (Equation 24) in the SIS model, we can obtain:(49)$$\left\{\begin{array}{c} \mu_{R_{0\max}}=1,\\ \mu_{R_{0\min}}=1. \end{array}\right.$$

Next, pressing the above results into formula (Equation 44), we can get $\mu_{S}=1$. Therefore, the SEIR model is identifiable. Namely, the parameter set $\Theta_{3}=\left\{D,W,R_{0\max},R_{0\min}\right\}$ can be uniquely determined from the observable *I*.

### 3.4. SEIRS Model

For the SEIRS model, we test whether the parameter set $\Theta_{4}=\left\{D,L,W,R_{0\max},R_{0\min}\right\}$ can be determined uniquely from observable *I*. First, we investigate whether the differential equation in Equation (Equation 4) can be decomposed into a sum of linearly independent functions. For the differential equation associated with *S*, we have:(50)$$\begin{array}{c} f_{S1}=\frac{N-S-E-I}{L},\;f_{S2}=-\frac{\beta (t)IS}{N}. \end{array}$$

According to Theorem A1, the generalized Wronskian determinant is as follows:(51)$$W_{S}=\left|\begin{array}{cc} {▵}_{0}f_{S1} & {▵}_{0}f_{S2}\\ {▵}_{1}f_{S1} & {▵}_{1}f_{S2} \end{array}\right|=\left|\begin{array}{cc} \frac{N-S-E-I}{L} & -\frac{\beta (t)IS}{N}\\ \frac{1}{L} & -\frac{\beta (t)(S+I)}{N} \end{array}\right|$$(52)$$\begin{array}{c} =\frac{\beta (t)}{NL}\left[IS-(N-S-E-I)(S+I)\right]\neq 0. \end{array}$$

So, $f_{S1}$ and $f_{S2}$ are linearly independent functions. Similarly, for the differential equation associated with *E*, we have:(53)$$\begin{array}{c} f_{E1}=\frac{\beta (t)IS}{N},\;f_{E2}=-\frac{E}{W}. \end{array}$$

The corresponding generalized Wronskian determinant is as follows:(54)$$W_{E}=\left|\begin{array}{cc} {▵}_{0}f_{E1} & {▵}_{0}f_{E2}\\ {▵}_{1}f_{E1} & {▵}_{1}f_{E2} \end{array}\right|=\left|\begin{array}{cc} \frac{\beta (t)IS}{N} & -\frac{E}{W}\\ \frac{\beta (t)(S+I)}{N} & -\frac{1}{W} \end{array}\right|=\frac{\beta (t)}{NW}\left[E(S+I)-IS\right]\neq 0.$$

So, $f_{E1}$ and $f_{E2}$ are linearly independent functions. For the differential equation associated with *I*, we have:(55)$$\begin{array}{c} f_{I1}=\frac{E}{W},\;f_{I2}=-\frac{I}{D}. \end{array}$$

The generalized Wronskian determinant is as follows:(56)$$W_{I}=\left|\begin{array}{cc} {▵}_{0}f_{I1} & {▵}_{0}f_{I2}\\ {▵}_{1}f_{I1} & {▵}_{1}f_{I2} \end{array}\right|=\left|\begin{array}{cc} \frac{E}{W} & -\frac{I}{D}\\ \frac{1}{W} & -\frac{1}{D} \end{array}\right|=\frac{I-E}{WD}\neq 0.$$

So, $f_{I1}$ and $f_{I2}$ are linearly independent functions. Next, we explore whether the parameter set $\Theta_{4}=\left\{D,L,W,R_{0\max},R_{0\min}\right\}$ can be determined uniquely from the observable *I* using the scaling method. The steps of the scaling method are as follows:Step 1.We scale the parameter set $\Theta_{4}=\left\{D,L,W,R_{0\max},R_{0\min}\right\}$ and the unobserved variable (*S*) by unknown scaling factors:
(57)$$\left\{\begin{array}{c} L\to \mu_{L}L,\\ W\to \mu_{W}W,\\ D\to \mu_{D}D,\\ R_{0\max}\to \mu_{R_{0\max}}R_{0\max},\\ R_{0\min}\to \mu_{R_{0\min}}R_{0\min},\\ S\to \mu_{S}S,\\ E\to \mu_{E}E. \end{array}\right.$$Step 2.We obtain the scaled version for each functional linear independent function in Equations (Equation 50), (Equation 53) and (Equation 55).Step 3.We obtain the identifiability equations:
(58)$$\frac{N-\mu_{S}S-\mu_{E}E-I}{\mu_{S}\mu_{L}L}=\frac{N-S-E-I}{L}.$$
(59)$$\begin{array}{cc} \; & \frac{I\mu_{S}S}{\mu_{S}N\mu_{D}D}\left[e^{aq(t)}\left(\mu_{R_{0\max}}R_{0\max}-\mu_{R_{0\min}}R_{0\min}\right)+\mu_{R_{0\min}}R_{0\min}\right]\\ \; & \;\;=\frac{IS}{ND}\left[e^{aq(t)}\left(R_{0\max}-R_{0\min}\right)+R_{0\min}\right], \end{array}$$
(60)$$\begin{array}{cc} \; & \frac{I\mu_{S}S}{\mu_{E}N\mu_{D}D}\left[e^{aq(t)}\left(\mu_{R_{0\max}}R_{0\max}-\mu_{R_{0\min}}R_{0\min}\right)+\mu_{R_{0\min}}R_{0\min}\right],\\ \; & \;\;=\frac{IS}{ND}\left[e^{aq(t)}\left(R_{0\max}-R_{0\min}\right)+R_{0\min}\right], \end{array}$$
(61)$$\begin{array}{c} \frac{\mu_{E}E}{\mu_{E}\mu_{W}W}=\frac{E}{W}, \end{array}$$
(62)$$\begin{array}{c} \frac{\mu_{E}E}{\mu_{W}W}=\frac{E}{W}, \end{array}$$
(63)$$\begin{array}{c} \frac{I}{\mu_{D}D}=\frac{I}{D}. \end{array}$$

Similar to the previous derivation of the corresponding part of the SEIR model, it is easy to obtain:(64)$$\left\{\begin{array}{c} \mu_{L}=1,\\ \mu_{W}=1,\\ \mu_{D}=1,\\ \mu_{R_{0\max}}=1,\\ \mu_{R_{0\min}}=1,\\ \mu_{S}=1,\\ \mu_{E}=1. \end{array}\right.$$

Therefore, the SEIRS model is identifiable. Namely, the parameter set $\Theta_{4}=\{D,L,W,R_{0\max},$$R_{0\min}\}$ can be determined uniquely from the observable *I*.

## 4. Discussion

In some existing studies, the authors explored the performance of different data assimilation algorithms when inferring parameters from observational epidemic data [13]. For example, ensemble filters are better at reproducing historical influenza incidence time series than particle Markov chain Monte Carlo. However, we still do not know whether the model is structurally unidentifiable, which affects the performance of the optimization algorithms. In this study, we applied the scaling method to analyse the structural identifiability of four types of commonly used humidity-driven epidemiological models when prevalence data is observable. Specifically, we investigated whether each parameter set $\Theta_{1}=\left\{D,R_{0\max},R_{0\min}\right\}$ in the SIS model, $\Theta_{2}=\left\{D,L,R_{0\max},R_{0\min}\right\}$ in the SIRS model, $\Theta_{3}=\left\{D,W,R_{0\max},R_{0\min}\right\}$ in the SEIR model and $\Theta_{4}=\left\{D,L,W,R_{0\max},R_{0\min}\right\}$ in the SEIRS model can be uniquely determined (*D* is the infectious period, *L* is the average duration of immunity, *W* is the latency period, and $R_{0\max},R_{0\min}$ are the combination of the maximum and minimum daily basic reproductive number and the minimum daily basic reproductive number). We find each model is identifiable when the prevalence is observable. Aside from considering prevalence data as observational data, this study also considered the number of new cases as observational data. For example, when we introduced the differential equation of the new case changes over time in the SIRS model, the differential equation decomposition components (obtained based on Equation (Equation 8) from the frameworks of the scaling method part) are not linear-independent. So the scaling method cannot be used to determine whether the model is structurally identifiable when the new cases are observable.

Much work needs to be done to further test, validate, and improve the ability of humidity-driven epidemiological models to predict influenza activity. In the future, we will test more expanded humidity-driven epidemiological models (such as a model that includes more than one exposed state) to provide theoretical support for these models to be used in practice. This investigation involved testing of the structure identifiability of different humidity-driven epidemiological models. In the future, we will assess the practical identifiability of these models to provide synthetic experimental support to enable these models to be used in practice. In addition, we will explore other model structure identifiability methods to test the structural identification of the humidity-driven epidemiological model when the number of new cases is observable. We will also consider more data types, such as the cumulative number of incidences.

In conclusion, our analysis suggests that the structural identifiability of these models can lay the foundation for testing practical identifiability in the future. In practice, epidemiological models should be examined with caution before using them to estimate model parameters from epidemic data.

## Figures and Tables

**Figure 1 viruses-14-02795-f001:**
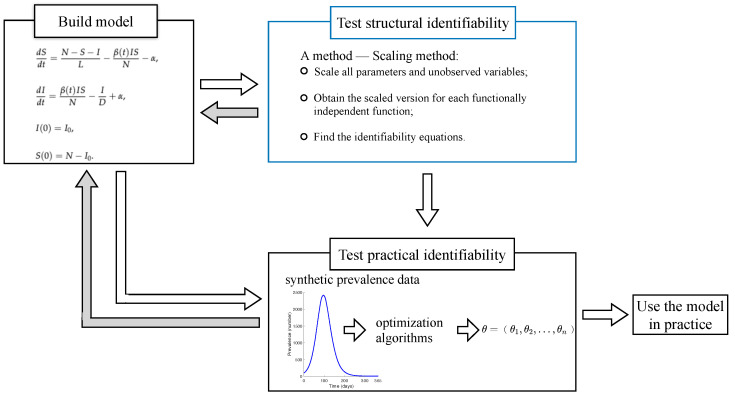
The process from building the model to putting it into use. The grey arrows show the feedback from testing the model identifiability results to modifying the model. The blue box illustrates the process of testing the model structural identifiability that our work focuses on.

## Data Availability

Not applicable.

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
