# Peer review of "The Structural Identifiability of a Humidity-Driven Epidemiological Model of Influenza Transmission"

_viruses, 2022, doi:10.3390/v14122795_

Round 1
Reviewer 1 Report
The author made a good try at applying the scaling method to analyze the structural identification of the humidity-driven epidemiological model. This is helpful for the filed although the novelty is not that big. I have no further questions right now.
Reviewer 2 Report
The research question is well defined, and the methods and results are accurate and thoroughly presented. The scope of the study is very limited -- it only shows structural identifiability for a single model, and does not address practical identifiability at all (although this is identified as future work). As a result, it is very brief even for a brief report.
The English could be edited to be clearer -- sometimes there are missing/extra words, perhaps because the sentences are very long. For example, lines 34-37 could be rewritten as "However, the model structure identifiability needs to be tested to avoid the optimization algorithm falling into a set of locally optimal solution when existing multiple sets of locally optimal solution in the search space without a global optimal solution." -- should this not be something like
"However, the model structure identifiability needs to be tested to avoid the optimization algorithm falling into a locally optimal solution rather than a global optimal solution. This can occur when there are multiple locally optimal solutions and a single global optimal solution."?
Does Case 2 not follow trivially from Case 1, since b = log ( R 0max − R 0min ) has a unique solution for R_0max given b and R_0min?
Also, Theorems A1 and A2 are not used and should be removed.
